# What Is the Impact of the Renewable Energy Power Absorption Guarantee Mechanism on China's Green Electricity Market?

Yan Lu [1], Xuan Liu [1], Hongjian Li [1], Haoran Wang [2,*], Jiajie Kong [2], Cheng Zhong [2], Mingli Cui [2], Yan Li [2], Xiaoqi Sun [2], Jiadong Xuan [2] and Tiantian Feng [2]

[1] State Grid Jibei Electric Power Company Limited Economic Research Institute, Beijing 100038, China; hdluyan@163.com (Y.L.); 18311035066@163.com (X.L.); 18601213209@126.com (H.L.)

[2] School of Economy and Management, China University of Geosciences Beijing, Beijing 100083, China; bs18xxkjj@126.com (J.K.); cheng4754105@163.com (C.Z.); cuimingli99@163.com (M.C.); 17801186166@163.com (Y.L.); sxq20001127@163.com (X.S.); xjd19811732636@163.com (J.X.); fengtiantian89@163.com (T.F.)

[*] Correspondence: 15732137196@163.com

**Abstract:** In order to accelerate the construction of a clean, low-carbon, safe and efficient energy system, China set the provincial weight of responsibility for renewable energy power consumption and established a renewable energy power absorption guarantee mechanism in 2019. As a market incentive policy, it has enduring effect on the low-carbon transformation of the power industry. Firstly, the operation mechanism of the renewable energy consumption guarantee mechanism is analyzed. The general framework, core elements and supporting measures are clarified. Secondly, a stock-flow diagram is constructed based on the system dynamics method. It contains the green electricity market sub-module, the green-certificate market sub-module and the excess power absorption market sub-module. Finally, multiple scenarios are set up to simulate the impact of the green-certificate market and excess power absorption market improvements on the installed capacity and tariff of China's green power market. The renewable energy guarantee mechanism is an effective means to promote the consumption of green electricity in China. In addition, in the short term the cost of electricity for users has increased, but in the long term the cost of electricity shows a fluctuating downward trend. This study provides theoretical references for the formulation of clean and low-carbon policy objectives in the power industry and the optimization of market mechanisms.

**Keywords:** electricity market; renewable energy consumption; market-operation-mechanism analysis; system dynamics

## 1. Introduction

At the general debate of the seventy-fifth United Nations General Assembly, China proposed that it would increase its national contribution and strive to achieve carbon peaking by 2030 and carbon neutrality by 2060 [1]. In order to achieve the dual-carbon goal, the government has vigorously promoted the development of renewable energy. As renewable energy enters into large-scale development, however, the problem of new energy power abandonment rate cannot be ignored, and it is important to establish a policy mechanism adapted to the entry of renewable energy into the power market. In 2019, the National Development and Reform Commission National Energy Board issued the Notice on the Establishment of a Guarantee Mechanism for the Consumption of Renewable Energy Electricity, in which it is requested that the market entity that bears the responsibility for the consumption of electricity in the power transaction should give priority to the completion of the consumption of renewable-energy-electricity power transactions, and to remind market players of their consumption responsibilities in the review of medium- and long-term contracts. The policy sets the weight of responsibility for renewable energy power

consumption for each provincial administrative region, and guarantees the level of renewable energy power consumption from two indicators: the weight of responsibility for total power consumption and the weight of responsibility for non-hydro power consumption. The market entity that bears the responsibility of consumption can fulfill the responsibility of consumption weight using three ways: actual consumption of renewable energy, purchase of excess consumption and purchase of tradable green certificates (TGCs) [2]. As a market incentive policy, the renewable energy power consumption guarantee mechanism can help promote the low-carbon transformation of the power industry and build a new type of power system.

Accelerating the construction of a new type of power system dominated by new energy sources is an important initiative to realize the "dual-carbon" goal, and the construction of a green power trading system in which power-using enterprises purchase green power from new energy power generators such as photovoltaic and wind power is an important link in the new type of power system. Green power trading is an independent trading variety set up under the framework of medium- and long-term trading, and the products are mainly on-grid energy from renewable energy power generators such as wind power and photovoltaic, which can be expanded to be a part of eligible hydro-power in the future. In September 2021, the Chinese government issued the Green Power Trading Pilot Program [3], and the green power trading guides the consumption of green power in a market-oriented way, and transmits the environmental value of green power to the consumers.

In order to encourage enterprises to increase their consumption of green electricity, in December 2021, the central economic work conference pointed out that new renewable energy and raw-material energy use is not included in the total-energy-consumption control [4]. Establishing a mechanism linking green power trading to the weight of responsibility for renewable energy consumption, whereby market-based users can complete the weight of responsibility for renewable energy consumption by purchasing green power, help to further implement the requirement that new renewable energy and raw-material energy use is not included in the total energy consumption control, as well as to further stimulate the potential for whole-society green power consumption [5]. In July 2023, the central deepening of the reform committee's second meeting proposed the promotion of dual control of energy consumption to dual control of carbon emissions [6], which promotes local policies in favor of renewable energy projects, further liberalizing the use of green power in enterprises to complete the restrictions on the responsibility of renewable energy consumption, and break the constraints on the use of renewable energy.

With the deepening of electricity market-oriented reforms, the weighting of responsibility for renewable electricity consumption has been successfully linked to green power trading. However, previous studies failed to focus on the impact of the renewable electricity consumption guarantee mechanisms on green electricity markets. Based on this, this paper attempts to handle the research keys as follows: How the renewable energy consumption guarantee mechanisms affect green electricity markets. In what ways and to what extent do renewable energy consumption guarantee mechanisms affect green electricity markets?

The main innovations and novelties of this research are divided into two aspects: Firstly, this study constructs a system dynamics model applicable to the renewable-energy-electricity consumption guarantee mechanism in the Beijing–Tianjin–Hebei region. Through the feedback of renewable-energy-electricity consumption policies on the electricity market, this study will promote the energy structure adjustment of the power industry. Secondly, this study will design a coupling mechanism between the green electricity market, green-certificate market, and excess-consumption market, and, through multiple scenarios, design and simulate the future market-reform measures, and explore effective measures to promote the consumption of renewable energy.

## 2. Literature Review

### 2.1. The Consumption of Renewable Energy in the World

In promoting the consumption of renewable energy, the systems of the United States, the European Union and other countries have been relatively perfect. Many countries have adopted RPS-type projects, including Germany, Sweden, Italy and Britain [7]. By 2019, 30 states and the District of Columbia had adopted the mandatory RPS system [8]. In 2018, the EU issued RED II (Renewable Energy Directive II), proposing that the proportion of new energy should reach at least 32% by 2030, and it will be increased to 45% by 2022, which greatly promoted the development and use of new energy [9]. Moreover, the EU has formed a GO (guarantee of origin) voluntary market for green certificates, and countries also have their own quotas (such as Sweden and Norway), electricity prices or premium policies (such as Britain and Germany) [10]. Europe uses the GO mechanism to encourage trading with GO between renewable-energy power-generation enterprises and power-purchase enterprises [11]. At present, the international renewable energy subsidy system mainly includes government-subsidized renewable energy incentive policy and market-oriented renewable energy incentive policy. The government-subsidized renewable energy incentive policy is represented by FIP policy and FIT policy, each of which has its own advantages and disadvantages. Japan introduced the FIT policy in 2012, which quickly popularized photovoltaic power generation in a short period of time. However, due to the large number of medium to old thermal power plants closed in the liberalized power market competition, it has also led to a shortage of power supply [12]. Winter and Schlesewsky pointed out that Germany's FIT electricity-price regulation promoted the development of renewable energy power, but at the same time it brought the problem of distorted distribution [13]. Du and Ma found that FIT can help Germany achieve the goal of encouraging the integration of the wind-energy market, but it hinders and delays the investment in solar energy technology [14]. FIP introduces a certain degree of market competition on the basis of FIT, which is a transitional way of renewable energy power generation from full acquisition to full bidding, which can effectively control costs and guide the sustainable development of renewable energy [15]. South Korea experienced a drastic transformation from FIT to RPS in 2012, and most renewable energy technologies were constantly updated and increased [16].

To address climate change, countries around the world have encouraged wind power, photovoltaic power, and hydroelectric power projects, and achieved significant effects [17–19]. The International Energy Agency predicts that global energy demand will be reduced by 10% by 2050, and renewable energy will then account for more than 2/3 of energy demand, with an increasing role for renewable energy sources [20]. Robalino López et al. proposed a systematic dynamic model of renewable energy and $CO_2$ emissions in the case of Ecuador, and found that with the growth of renewable energy, the improvement of the structure of the production sector, and the use of more efficient fossil fuel technologies, it is possible to control $CO_2$ emissions in the future [21]. Liu et al. showed that renewable energy consumption is a two-way Granger because of GDP, i.e., renewable energy consumption and economic growth can promote each other [22]. Li et al. pointed out that the literature also points out that renewable energy output is characterized by randomness, volatility and difficulty in forecasting. The instability and transition cost of renewable energy is also a challenge for its subsequent development and application [23].

### 2.2. Effects of China's Renewable Energy Consumption Guarantee Mechanism

The Renewable Energy Consumption Guarantee Mechanism is essentially an RPS (Renewable Portfolio Standards) policy with Chinese characteristics, and the research methods for its implementation effects are mainly divided into system simulation and econometric methods.

In terms of system simulation, scholars are concerned about the policy's possible impacts on the power system and its synergistic relationship with other low-carbon energy policies. Yan et al. simulated the synergistic relationship between the renewable energy con-

sumption guarantee mechanism and the power market by constructing a system dynamics model of the consumption-quota market and the renewable energy power market based on the power data of Yunnan Province, and analyzed the implementation effect of different consumption-quota schemes [24]. Xu et al. analyzed the linkage characteristics and different influencing factors of trading price between China's green-certificate trading market and power market. They have constructed a green-certificate price-mechanism model under the framework of the renewable energy consumption guarantee mechanism [25]. Guo predicted the implementation pressure of each province of China by constructing a composite-prediction model, and used the game equilibrium theory to calculate the equilibrium price of the consumption-quota trading, and finally assessed the potential of renewable energy development in typical provinces and the impact of policy on renewable energy consumption [26]. Song et al. established a system dynamics model for the renewable energy consumption guarantee mechanism and the green-power-certificate trading market to compare the effects of different policy combinations on multiple markets and renewable energy capacity [27]. Zhu et al. analyze the impacts of the Renewable Portfolio Standards on the retail electricity market and the proposed policy improvement suggestions by constructing a three-party evolutionary game system dynamics model that includes the regulator and two types of power-sales companies with different strengths [28]. Fang et al. constructed a system dynamics model between the regulator, renewable energy power generators and thermal power generators to examine the implementation effects of the guarantee mechanism on the electricity sales side from the perspective of cost structure [29]. Wang and Li considered the green-certificate market and the electricity market, and used game theory to construct a bi-level decision-making model with government and different power generators in order to simulate the emission reduction effect and renewable energy consumption following the guarantee mechanism's implementation [30].

Studies using econometrics methodology have analyzed the effect of the guarantee mechanism based on long-term power-statistics data during its implementation period. Shrimali and Kniefel measured the promotion effect of RPS policy on renewable energy using panel regression models. They used data from 1991–2007 for each state in the U.S. and found that the RPS policy, as a key factor, has a significant effect on all types of renewable energy penetration increase, which has a significant impact [31]. Lee and Seo measured the long-term implementation effect of RPS policy in South Korea based on the historical data using LGM (Latent Growth Model) and found that its effectiveness may be decreasing linearly from year to year [32]. Zhou and Solomon considered a wide range of key variables based on panel data from 1998–2017 using mixed random effects negative binomial regression to study the impact of RPS on renewable energy capacity in the USA, and found that RPS has significantly promoted the development of renewable energy in each state [33]. Joshi utilized generalized least squares with a panel-corrected standard-errors model to measure the promotion effect of the RPS policy on renewable energy development using data from 1990–2014, and found that RPS had promoted more than 33% of the growth of renewable energy capacity, with inconsistent effects on different power sources [34].

In summary, due to the lack of historical data because of the short time of policy implementation, scholars in China tend to choose the simulation approach to study the implementation effect of the guarantee mechanism [24–30]. In contrast, for countries with a longer period of renewable energy development, such as the USA and the UK, scholars mainly utilize econometric methods to assess the implementation effects of the RPS system based on long-term data [31–34].

## 2.3. Implementation Effects of Low-Carbon Policies

As important means to realize carbon emission reduction goals, low-carbon policies such as carbon emissions trading policy, green-power-certificate trading policy and green power trading policy have been widely evaluated in recent years [35–38]. For the effectiveness of their implementation effects, scholars mainly evaluate them by constructing

econometric models and equilibrium models. The econometric difference method is often used to test the effectiveness of carbon-emissions trading policies. For example, Zhang et al. used the difference-in-difference (DID) method to evaluate the impact of the implementation of a carbon-emissions trading system (ETS) based on industrial carbon-emissions data from 30 provinces in China from 2008 to 2016, and the result showed that the implementation of the carbon trading policy significantly reduced the industrial $CO_2$ emissions (24.2%) in seven pilot cities [39]. In addition, Gao et al. further utilized the DID method to evaluate the effectiveness of ETS. The conclusion shows that ETS not only helps to mitigate carbon emissions in pilot regions and industries, but its role in reducing production-based emissions is greater than the impact of consumption-based emissions [40]. In terms of general equilibrium models, Lin et al. constructed a dynamic recursive computable general equilibrium model to study the economic, energy and environmental impacts of a national carbon-emissions trading market [41]. Helgesen et al. developed a hybrid complementary, multi-regional, partial equilibrium market clearing model for electricity- and green power certificates under the assumption of competition in the Nash-Guno market in order to investigate the economic impact of tradable green certificates (TGCs) on promoting renewable energy generation [42]. An et al. proposed a two-stage joint-equilibrium model based on the theory of monopolistic competitive equilibrium to quantitatively study the impact of the introduction of green-power-certificate policy on the wholesale-electricity market [43].

In addition to researching on the effects of individual low-carbon policies, scholars have also conducted extensive research on the effects of synergistic implementation between different low-carbon policies. Existing research methods for the synergistic development of different policies are mainly based on synergy theory using system dynamics or mathematical modeling. In the research on the synergistic effect of green-certificate trading policy and carbon-emissions trading policy, system dynamics models of green-certificate trading policy and carbon-emissions trading policy were constructed to explore their interrelationships and mechanisms of action. Liu et al. concluded that the parallel implementation of green certificates and a carbon-emissions trading system can optimize the power structure and carbon-emission reduction [44], while Tan et al. concluded that the synergistic effect of the two policies on the pricing of feed-in tariffs and the structure of the power-generation industry is positive [45]. In the research of multiple policies' synergies, Wei et al. constructed an interactive model of a carbon trading market and green power policy, exploring the emission reduction effect in green power policy, green-certificate policy and the interactive mechanism of the transmission of quota price in the carbon market [46]. Ge et al. set up a multi-regional, multi-market equilibrium model to analyze the coupling relationship between China's carbon trading market and the green-certificate trading with the electric power market, and emphasized the advantages and effectiveness of the market coupling [47]. Feng et al. established and simulated a coupling model between the green certificates, carbon trading and electricity market in China, as well as a policy-synergy model between carbon trading and green-certificate trading. The key points of coupling and synergy between different policies and between different markets were proposed and optimization suggestions for the co-implementation of carbon trading, green-certificate trading and a variety of carbon emission reduction policies were provided in their study [48].

## 3. Operation Mechanism of Renewable-Energy-Electricity Consumption Guarantee Mechanism

The world is currently recovering from the COVID-19 and is gradually taking steps to address the energy crisis. Global energy investment rebounded by nearly 10 percent in 2022. According to the IEA, this recovery is driven by a strong growth in renewable energy investment. According to the IEA's 26 May 2023 report, World Energy Investment 2023, it can be seen that clean-energy investment will grow at a rate of 24% in 2023, much faster than the 15% growth rate in fossil fuel investment. For every dollar invested in fossil fuels, $1.7 is invested in clean energy, compared to a ratio of 1:1 five years ago.

Key areas of investment include renewable energy, nuclear energy, power grids, energy storage, low-emission fuels, energy-efficiency improvements, and electrification. As more countries closely align climate- and energy-security objectives, policy support for clean energy has been strengthened, seeking to achieve a cleaner, low-carbon energy transition while strengthening their own industrial capacity.

In China, wind power, photovoltaic power generation as the representative of renewable energy investment and construction has been effective. The scale of installed power has steadily ranked first in the world. The share of power generation has steadily increased; the cost is rapidly declining, and has basically entered a new stage of independent and sustainable development. In 2022, the national wind power and photovoltaic power generator's new installed capacity exceeded 120 million kilowatts. For three consecutive years, it exceeded 100 million kilowatts, hitting a new record high. Annual renewable energy's new installed capacity was 152 million kilowatts, accounting for 76.2% of the country's new power generation installed capacity, and has become the main body of China's new power installed capacity.

However, with the continuous growth of China's new energy installed capacity, the phenomenon of wind and light abandonment has occurred in localized areas at the same time, in order to accelerate the planning and construction of a new type of energy system, and to promote the consumption of renewable energy and high-quality development. In May 2019, the National Development and Reform Commission (NDRC) and the National Energy Administration (NEA) jointly issued the "Notice on Establishing and Improving the Renewable Energy Electricity Consumption Guarantee Mechanism" and the "Responsibility Weights for the Total Consumption of Renewable Energy and Non hydroelectric Renewable Energy Electricity in Various Provinces (Autonomous Regions, Municipalities)". The release of the notice indicates that the Chinese government has formally established an assessment system for the consumption of renewable energy.

The consumption guarantee mechanism applies the market-oriented way to achieve the optimization of renewable energy across the provincial and regional large-scale allocations, and to ensure the renewable energy power market consumption. The consumption guarantee mechanism in China is divided by provincial administrative regions, and is mainly divided into two categories in terms of the obligation subjects responsible for the consumption of renewable energy electricity. One type are power grid enterprises that directly supply and sell electricity to users, independent power sellers, and power sellers with power-distribution-network operation rights; the other type is electricity users who purchase electricity through the wholesale-electricity market and enterprises that own their own power plants [49].

In the renewable-energy-electricity consumption volume trading market, it is stipulated that if the market entity responsible for consumption fails to complete the consumption volume corresponding to the weight of the consumption responsibility, they can choose to purchase from the over-completion in the market of the excess-consumption volume or participate in the market of green-certificate markets to subscribe for the completion of the two ways, and the green certificates corresponding to the equivalent amount of renewable energy power is credited as the amount of consumption volume. The implementation of the renewable-energy-electricity consumption guarantee mechanism has led to closer spatiotemporal scale interactions among multiple markets such as the electricity market, consumption-volume market, and green-certificate market. The operation mechanism of the renewable energy consumption guarantee mechanism is essentially a multi-level market transaction centered on electricity, as shown in Figure 1. Its basic composition includes three markets, as well as the influence of relationship and dynamic cycles of resources between multiple market entities. The blue arrow represents the flow of electricity, the green arrow represents the flow of green certificates, and the yellow arrow represents the flow of excess consumption.

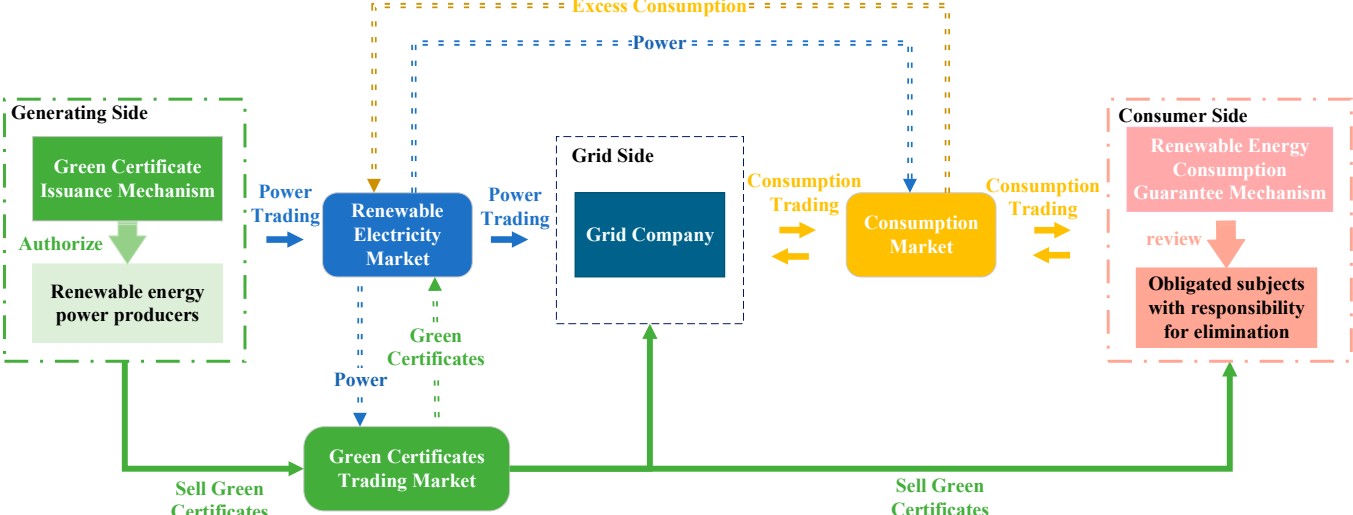

**Figure 1.** Operating mechanism of renewable-energy-electricity consumption guarantee mechanism.

Specifically, renewable energy generators are suppliers to the consumption market and electricity market, selling electricity and consumption to market entities. Market entities can sell excess consumption to generate income and purchase the insufficient portion of their consumption obligations in the consumption-volume market. The increase in electricity generation has affected the supply of electricity and green certificates, thereby reducing the prices of renewable-energy-electricity prices and green certificates, making it easier for market entities to fulfill their consumption obligations by purchasing renewable energy electricity and green certificates. The excess consumption is sold in the market, thereby increasing the market supply of consumption and lowering the price of consumption. At the same time, fluctuations in electricity prices often affect changes in the income of power producers, prompting them to increase the production of renewable energy, which will further adjust the supply-and-demand structure of green certificates and consumption.

## 4. Materials and Methods

### 4.1. System Boundary and Model Assumptions

Since system dynamics (SD) is a subject that studies the dynamic complexity of a system; it is mainly used to study the structure and function of a complex system and the interdependence of dynamic behaviors. This method can analyze the causal relationship and feedback mechanism among decision-making units from a micro perspective, and can predict the future dynamic behaviors. Therefore, this section uses the SD model to stimulate the impact of the renewable energy consumption guarantee mechanism on the renewable-energy-electricity market.

System definition: This paper will study the interaction mechanism between the RES-E consumption guarantee mechanism and the electricity market. The mechanism of the excess-consumption trading market, green-certificate trading (TGC) market and electricity market is studied. In this part, the system dynamics (SD) model is composed of an excess-consumption trading market subsystem, TGC market subsystem and power market subsystem.

**Hypothesis 1. (H1).** *The market players who bear the responsibility of consumption mainly complete the responsibility of consumption by actually consuming renewable energy power, and the insufficient part is completed by purchasing excess consumption or a green certificate.*

**Hypothesis 2. (H2).** *In the excess-consumption market, the transaction subject of excess consumption is the power user, which is divided into the seller and the buyer. In the system dynamics model system constructed in this report, excess-consumption transactions take precedence over TGC transactions.*

**Hypothesis 3. (H3).** *In the TGC market, the supplier of green certificates is renewable energy power generation companies, and the demand for green certificates is electricity retailers, power-grid companies, power users, and enterprises with self-owned power plants. Assuming that 1 MWh power can be redeemed for 1 unit of green certificate, a green certificate is issued once a month.*

**Hypothesis 4. (H4).** *The electricity suppliers in the electricity market only consider the renewable energy generation. The types of renewable energy generation include wind power generation, photovoltaic power generation and biomass power generation. The transaction of renewable energy power is separated from the transaction of green certificates, and there is no bundled transaction.*

### 4.2. System Dynamics Model Based on Consumption Market and Renewable Energy Market

Based on the system structure and modeling principles of system dynamics, the interaction model between the RES-E consumption guarantee mechanism and the electricity market is drawn using Vensim PLE 5.7.2.1 simulation software. It includes two modules: a renewable energy generation electronic subsystem and a renewable energy power consumption subsystem. The obtained stock flow chart is shown in Figure 2. Both subsystems determine the relationship between prices and supply and demand based on the supply and demand principles in microeconomics. Using different symbols to represent different types of variables, different types of arrows represent different functional relationships. Among them, INTEG, DELAY1 and SMOOTH are the integration function, delay function, and smoothing function in Vensim PLE, respectively. The variables in the model are divided into four categories: state variables, rate variables, auxiliary variables and constants. The single line arrow connecting state variables and auxiliary variables, double line arrow connecting rate variable. The model constructed in this study is based on the Beijing Tianjin Hebei region of China and has reference value for the renewable-energy-electricity consumption market in other provinces in China.

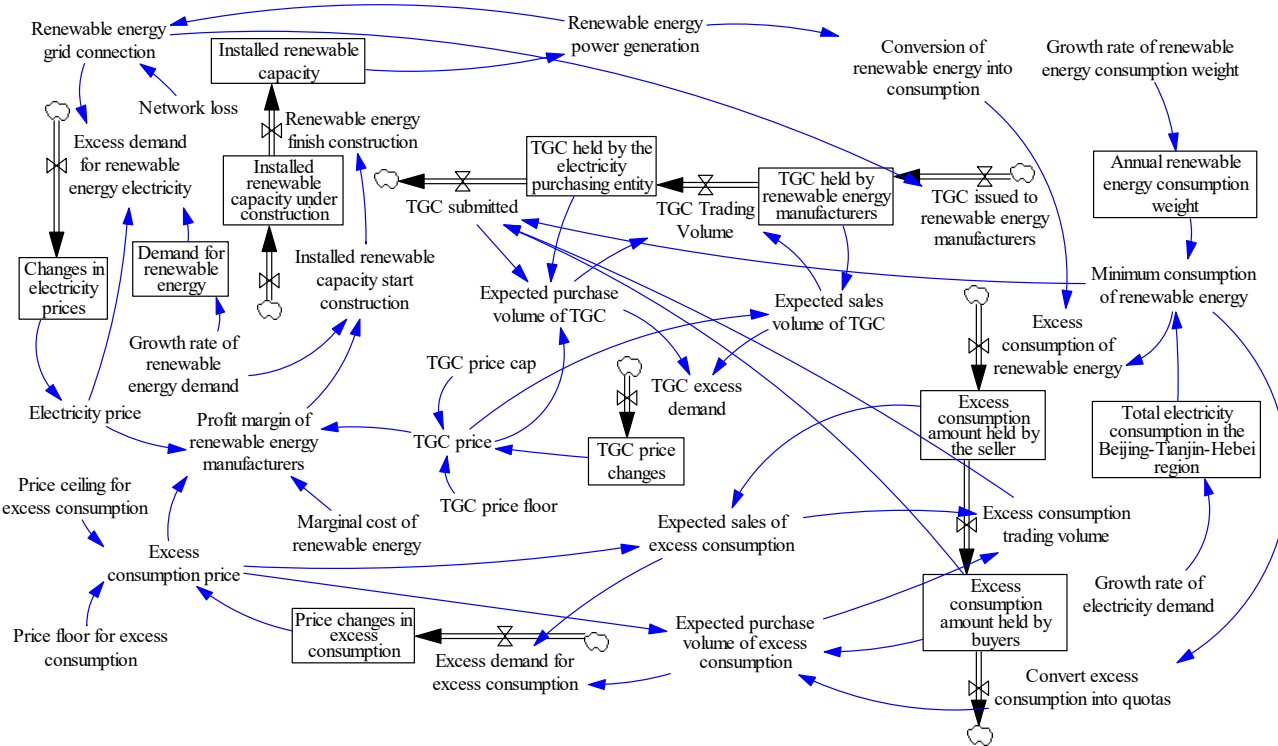

**Figure 2.** Stock flow chart.

### 4.3. System Dynamics Model Construction

In this section, we present the important equations of the model. The main function of the renewable energy market can be expressed as Equations (1)–(5):

$$B_r = P_p + P_t + P_{ec} - P_r \tag{1}$$

$$Q_{rb} = Q_{rb0} \times (1+r) \div 12 \text{ months} \div U \times (P_p + P_t + P_{ec} - P_r) \tag{2}$$

$$Q_{rf} = \text{DELAY1}(Q_{rb}, 12 \text{ months}, 0) \tag{3}$$

$$Q_{rz} = \text{INTEG}\left(Q_{rf}\right) \tag{4}$$

$$Q_r = Q_{rz} \times U \div 12 \text{ months} \tag{5}$$

where $B_r$ is profit margin renewable energy manufacturers; $Q_{rb}$ is installed renewable capacity start construction; $r$ is growth rate of renewable energy demand; $P_p$ is electricity price; $P_t$ is TGC price; $P_{ro}$ is excess-consumption price; $P_r$ is marginal cost of renewable energy; $Q_{rf}$ is installed renewable capacity under construction; $U$ is annual utilization hours of the unit; $Q_{rz}$ is installed renewable capacity; and $Q_r$ is renewable energy power generation.

The main function of renewable energy excess-consumption market (including green certificate) can be expressed as Equations (6)–(14):

$$V_{pt} = \text{INTEG}(q_{to}) \tag{6}$$

$$q_{tpb} = \text{SMOOTH}\begin{cases} 0; & q_{tb} > q_g \\ 0.16 \div p_t \times (q_g - q_{tb}); & q_{tb} \leq q_g \end{cases} \tag{7}$$

$$q_{tps} = p_t \times q_r \tag{8}$$

$$q_{to} = \text{SMOOTH}\begin{cases} -0.05; & q_{tpb} - q_{tps} < -0.05 \\ q_{tpb} - q_{tps}; & -0.05 \leq q_{tpb} - q_{tps} \leq 0.05 \\ 0.05; & q_{tpb} - q_{tps} > 0.05 \end{cases} \tag{9}$$

$$q_g = \text{INTEG}(EC_m - EC_b - EC_{bu}) \tag{10}$$

where $V_{pt}$ is TGC price changes; $q_{to}$ is TGC excess demand; $q_{tpb}$ is expected purchase volume of TGC; $q_{tb}$ is TGC held by the electricity purchasing entity; $q_g$ is TGC submitted; $q_{tps}$ is expected sales volume of TGC; $p_t$ is TGC price; $q_r$ is TGC held by renewable energy manufacturers; $q_{to}$ is TGC excess demand; $EC_m$ is minimum consumption of renewable energy; $EC_b$ is excess-consumption amount held by buyers; and $EC_{bu}$ is excess-consumption trading volume.

$$V_{pec} = \text{INTEG}(q_{eco}) \tag{11}$$

$$q_{ecpb} = \text{SMOOTH}\begin{cases} 0; & q_{ecb} > q_{ecg} \\ 0.06 \div p_{ec} \times (q_{ecg} - q_{ecb}); & q_{ecb} \leq q_{ecg} \end{cases} \tag{12}$$

$$q_{ecps} = p_{ec} \times q_{ecr} \tag{13}$$

$$q_{eco} = \text{SMOOTH}\begin{cases} -0.05; & q_{ecpb} - q_{ecps} < -0.05 \\ q_{ecpb} - q_{ecps}; & -0.05 \leq q_{ecpb} - q_{ecps} \leq 0.05 \\ 0.05; & q_{ecpb} - q_{ecps} > 0.05 \end{cases} \tag{14}$$

where $V_{pec}$ is TGC price changes; $q_{eco}$ is excess demand for excess consumption; $q_{ecpb}$ is expected purchase volume of excess consumption; $q_{ecb}$ is excess-consumption amount held by buyers; $q_{ecg}$ is convert excess consumption into quotas; $q_{ecps}$ is expected sales volume of excess consumption; $p_{ec}$ is excess-consumption price; $q_{ecr}$ is excess-consumption amount held by the seller; and $q_{eco}$ is excess demand for excess consumption.

### 4.4. Data Source and Related Parameter Description

The model takes 2021 as the base period, with a time step of 1 month, and a simulation time of 120 months (10 years) is set, that is, 2021–2030. The data on power generation, installed capacity, and power-generation costs are sourced from the China Statistical Yearbook and the power industry database on the China Electricity Union website. Referring to the historical data of China's green-power-certificate subscription trading platform, the initial price of the green certificate is set at 0.3 yuan/kWh. According to the Notice on the Responsibility Weights and Related Matters of Renewable Energy Electricity Consumption in 2021 issued by the National Development and Reform Commission, the annual growth rate of renewable energy consumption weight in the baseline scenario is 2.5%.

## 5. Simulation Results Analysis

### 5.1. Multi-Scenario Setting

In order to explore the effective improvement measures of the renewable energy power consumption market, the impact of market mechanism improvement is further studied. In this study, three scenarios were set up, namely the baseline scenario (BAU), the renewable energy power consumption weight scenario (REQ), the penalty mechanism scenario (PEM) and the comprehensive scenario (COM). Improving the power structure is an important path to achieve the goal of " double carbon" and the government will continue to strengthen the binding of renewable energy policies.

Firstly, increasing the weight of renewable energy consumption is one of the ways to enhance environmental constraints. Due to the increase in the proportion of renewable energy that users need to consume, it will promote the transformation of renewable energy, so the REQ scenarios have been set. Secondly, the punishment mechanism is also one of the means to enhance environmental constraints. The PEM scenarios were set up to increase the enthusiasm for consuming green electricity by punishing the profit of the consumer. Finally, future-policy-reform measures may be implemented in parallel, thus setting a COM scenario. The details are shown in Table 1.

**Table 1.** Multi-scenario setting.

| Scenario Setting | Scenario Number | Scenario Description |
|---|---|---|
| Baseline scenario | BAU | It is assumed that the simulation analysis is based on the existing renewable energy consumption guarantee mechanism. |
| Renewable energy power consumption weight | REQ1 | In 2030, the minimum renewable energy power consumption weight is 30%. |
| | REQ2 | In 2030, the minimum renewable energy power consumption weight is 40%. |
| Punishment mechanism | PEM1 | Punish the subject who fails to complete the task of consumption, 0.3 yuan/kWh. |
| | PEM2 | Punish the subject who fails to complete the task of consumption, 0.6 yuan/kWh. |
| Comprehensive scenario | COM | In 2030, the minimum renewable energy power consumption weight is 40%. Meanwhile, punish the subject who fails to complete the task of consumption, 0.6 yuan/kWh. |

## 5.2. The Impact of Renewable Energy Consumption Weight on the Market of Renewable Energy Consumption

Figure 3 shows the trend of green-certificate price under different renewable energy consumption weight scenarios. The figure takes 2021 as the base period and simulates and predicts the next 120 months (10 years). The curves of red, blue and green represent the prices of green certificates in the three scenarios of renewable energy power consumption weight of 30%, 40% and existing renewable energy consumption guarantee mechanism.

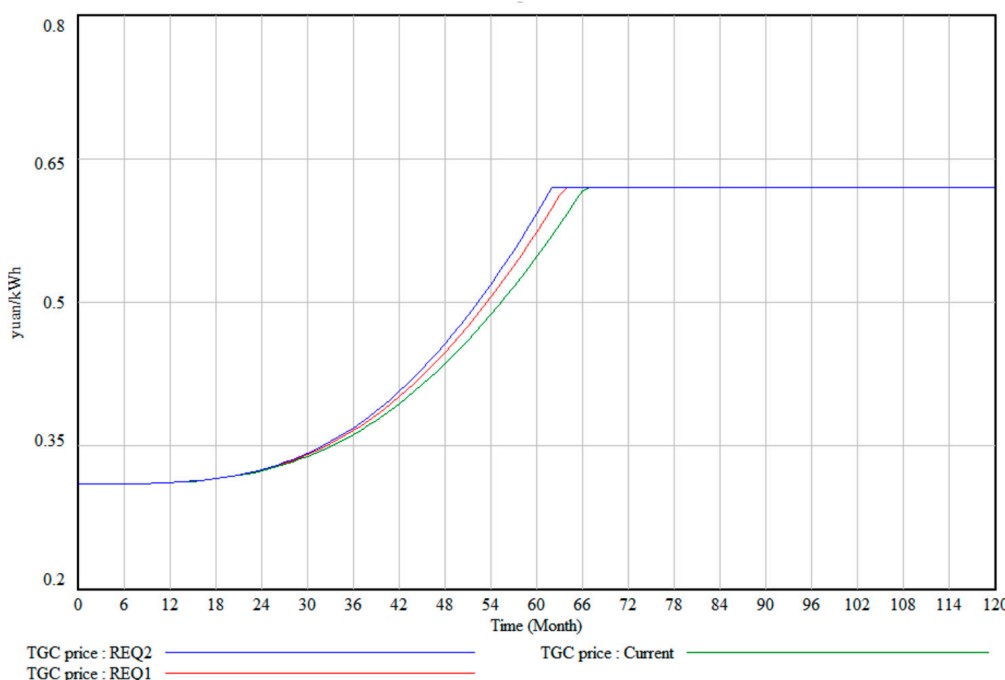

**Figure 3.** Green-certificate price under different renewable energy consumption weights.

It can be found that the price of a green certificate has experienced the process of 'low maintenance-rapid rise-high maintenance'. Specific analysis: With the increase in consumption weight, the price of a green certificate rises in advance. At the same time, the higher the minimum renewable energy consumption weight, the higher the price of the green certificate. The reason is that the higher the proportion of consumption, the more the demand for green certificates by relevant market players, thereby increasing the price of green certificates.

Figure 4 reflects the excess-consumption price under different renewable energy consumption weights. It can be seen from the figure that in the next 10 years, the price of excess consumption in the three scenarios will show a trend of rising first and then stabilizing. Specifically, the higher the minimum renewable energy consumption weight at the same time, the higher the price of the green certificate. That is, the price under the consumption ratio of 40% is higher than that under the consumption ratio of 30%, which is higher than the price under the existing mechanism, but the difference of the curve is not large. This is because the market players who bear the responsibility of consumption need to complete the responsibility of consumption through the actual consumption of renewable energy power, and the insufficient part can be completed by purchasing excess consumption. Therefore, a higher proportion of consumption leads to the relevant market players need to purchase excess consumption in order to complete the performance task, thus increasing the price of excess consumption.

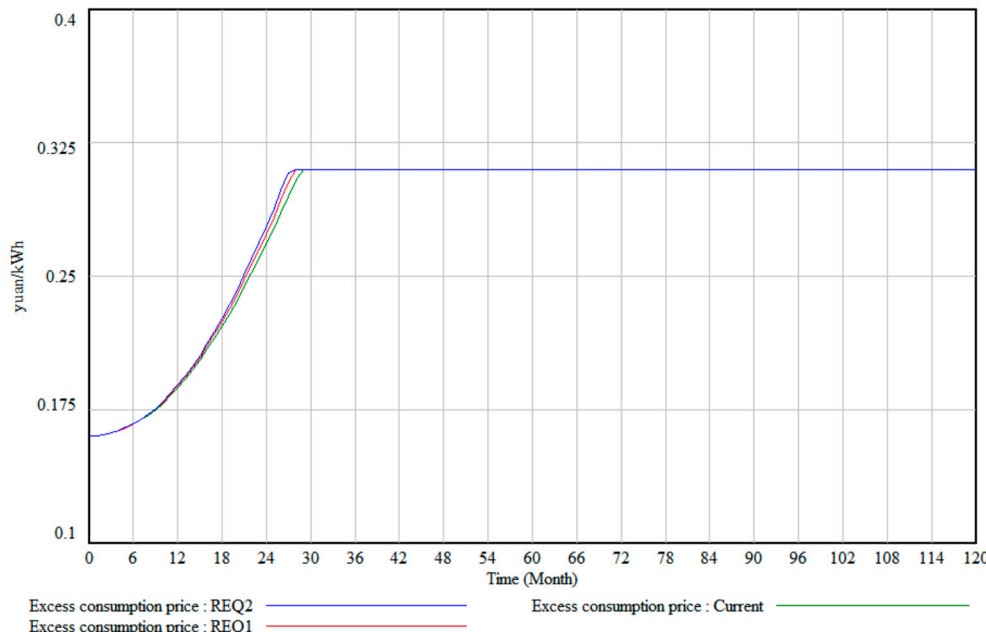

**Figure 4.** Excess-consumption price under different renewable energy consumption weights.

Figure 5 reflects the change of renewable energy's installed capacity under different renewable energy consumption weights. It can be seen from the figure that in the next 10 years, the installed capacity of renewable energy in the three scenarios shows a rising trend, but the difference between multiple scenarios is not significant. In the future, with the increase in carbon constraints, the cost of thermal power generation will increase, which will slow down or even reduce the growth of thermal power generation. Under the background of the increase in total social power demand, the power generation of renewable energy will increase rapidly. It can be seen from Figure 3 that the reason for the growth of renewable energy power does not come from the weight factor of renewable energy consumption.

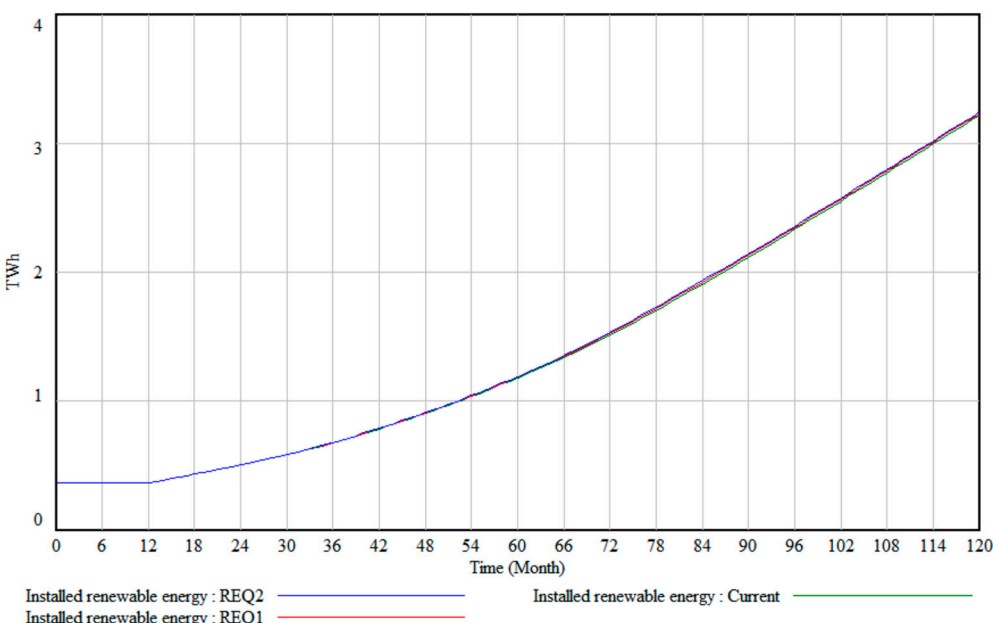

**Figure 5.** Renewable energy installation under different renewable energy consumption weights.

We found that it is not necessarily that the higher the weight, the greater the installed capacity of renewable energy from Figures 3–5. Notably, on the one hand, due to the limitation of installed capacity in the region, the marginal contribution of this effect is decreasing. On the other hand, due to the difference in the degree of stimulation of renewable-energy-electricity and green-certificate trading caused by the increase in weight, it may lead to the accumulation of more green certificates or excess consumption that have not been traded in the region. Given this, it is necessary to timely promote the integration of green certificates or excess consumption into the national unified market process.

### 5.3. Impact of Penalty Price on the Market of Renewable Energy Consumption

Figure 4 reflects the price change trend of green certificates under different penalty price scenarios. It can be seen from Figure 6 that in the next 10 years, the price of green certificates in the three scenarios show a trend of 'first maintaining stability, rising, then stabilizing'. In the early stage, the price of green certificates remained at a low level, and there was basically no difference in price in the three situations (about 0.30 yuan/kWh). In the following stage (about 6–48 months), the three curves all show an upward trend; that is, the price of green certificates is increasing, and the heavier the punishment implemented at the same time, the higher the price of green certificates, and the price difference is gradually widening. At this stage, higher punishment means that the market that bears the responsibility of consumption will certify more green certificates to fulfill the contract and avoid punishment. The higher the demand for green certificates will be, the higher the price of green certificates will be, according to the relationship between supply and demand. But in the end, the three curves tend to be stable.

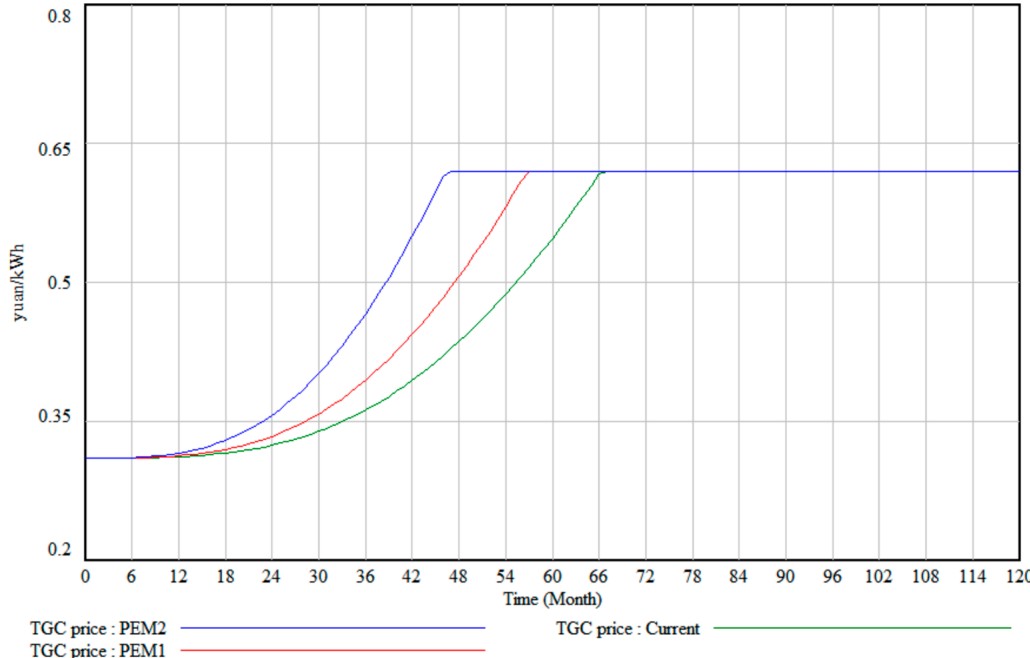

**Figure 6.** Green-certificate price under different penalty prices.

Figure 7 reflects the price of excess consumption under different penalty prices. It can be seen from Figure 7 that in the next 10 years, the prices of excess consumption in the three scenarios all show a trend of rising first and then stabilizing. In the first stage (about 0–28 months), the three curves are on the rise, that is, the price of excess consumption is increasing, from about 0.150 yuan/kWh to 0.300 yuan/kWh. The heavier the punishment at the same time during this period, the higher the price of excess consumption. With higher punishment, the market players who fail to complete the consumption task will complete the performance by purchasing excess consumption in order to avoid punishment, and

the demand for excess consumption will increase, and so the price of excess consumption will be higher. In the second stage (about 28 months–120 months), the price of excess consumption in the three scenarios remained stable after reaching about 0.300 yuan/kWh. It can be seen that the difference in penalty price will affect the price of excess consumption, but the final price will be stable at the same level.

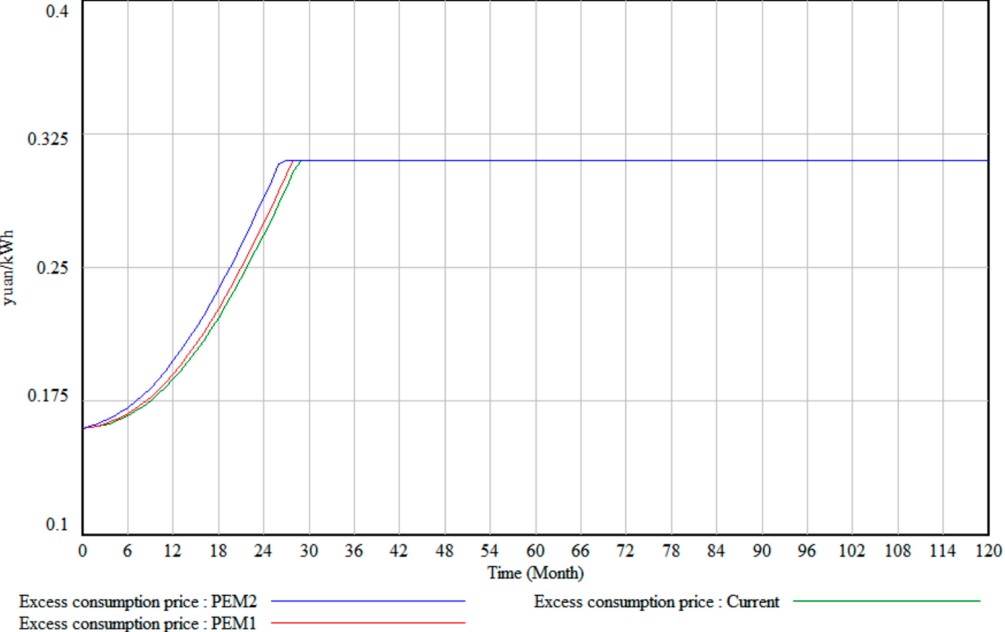

**Figure 7.** Excess-consumption price under different penalty prices.

Figure 8 reflects the installed capacity of renewable energy under different penalty price scenarios. It can be seen from Figure 8 that in the next 10 years, the installed capacity of renewable energy in the three scenarios will show a trend of stabilizing first and then gradually increasing. In the first stage (about 0–12 months), the three curves coincide, and the installed capacity of renewable energy in the three scenarios is maintained at about 0.4 billion kW, which indicates that the effect of different scenarios on the installed capacity is not obvious, and the effect of the punishment mechanism needs to be revealed after a certain period of time. Subsequently, in the two stages (12–120 months), the three curves continue to rise, and the installed capacity of renewable energy gradually increases. During the 12–36 months period, the three curves basically coincide, and the installed capacity of renewable energy in the three scenarios is basically the same. During the period of 36–120 months, the three curves are gradually separated. The heavier the penalty imposed at the same time, the higher the installed capacity of renewable energy. That is, the installed capacity of renewable energy under the penalty of 0.6 yuan/kWh is more than that of 0.3 yuan/kWh, which is higher than the price under the existing mechanism. The increase in renewable energy installation is mainly due to the punishment mechanism. In order to avoid punishment, the consumption amount corresponding to the completion of consumption responsibility will increase the demand for renewable energy power, which will promote renewable energy power producers to increase the output of renewable energy.

We found that punitive measures have added mandatory constraints to the market mechanism of renewable energy consumption policies, which is one of the effective means to stimulate consumption entities from Figures 6–8. Although there are currently no enterprises that have completed the weight indicators in this policy, there are no punitive measures. However, punitive measures need to be introduced in the future to ensure the effectiveness of renewable energy consumption policies.

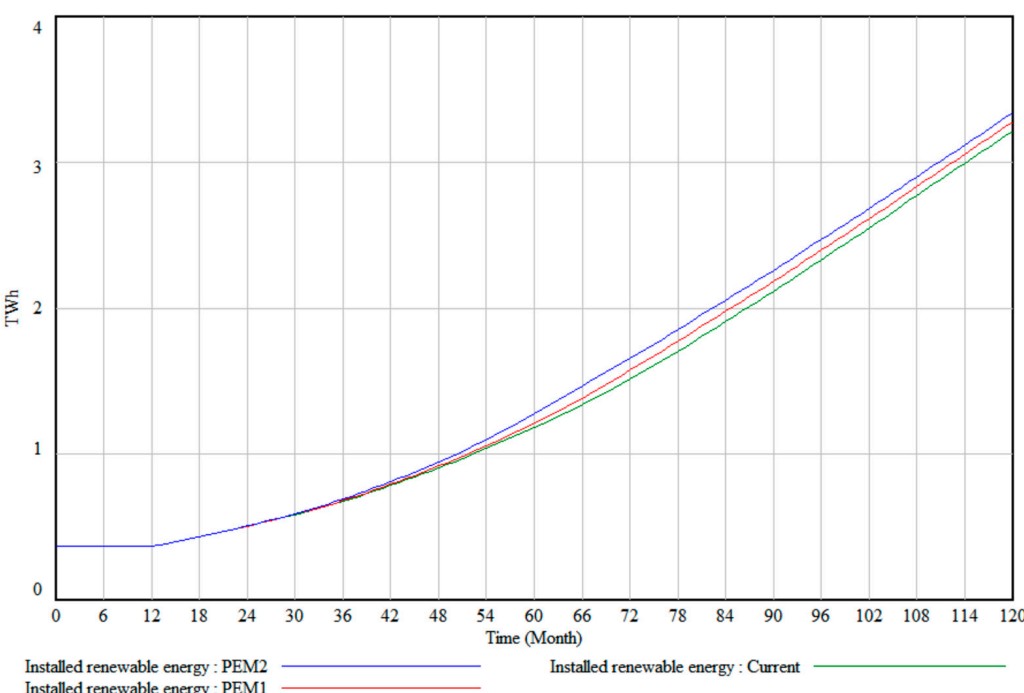

**Figure 8.** Renewable energy installation under different penalty prices.

*5.4. Impact of Integrated Policy Scenarios on Renewable Energy Consumption Market*

Figure 9 reflects the changes in the profit margins of renewable energy manufacturers in various scenarios. It can be seen from Figure 9 that the profit–space curve of renewable energy manufacturers is the highest in the comprehensive scenario. The increase in the weight of renewable energy has increased the demand for renewable energy by the subject of consumption responsibility. The introduction of the punishment mechanism prompts the subject of consumption responsibility to need to pay additional costs, which increases the binding of the subject of consumption responsibility to consumption. On the whole, no matter what kind of policy is implemented, it can promote the increase in profit space of renewable energy manufacturers.

Figure 10 reflects the changes of renewable energy power generation in the comprehensive scenario considering the increase in renewable energy consumption weight and penalty price, as well as in other scenarios. The renewable energy power generation in various scenarios shows an overall upward trend. Under the incentive of clean and low-carbon policies, the rise of green power generation has become an inevitable trend. Specifically, at the initial stage of planning (about 0–12 months), the curve overlap of each scenario remained at the same stable level (about 3 kw), and the effects of various policies during this period were not reflected. In the subsequent 12–120 months, the renewable energy power generation in various scenarios will increase, and the difference is gradually becoming increasingly evident, which can show that the effect of implementing comprehensive policies at the same time is better.

We found that both the consumption weight and punishment mechanism promote the means of renewable energy consumption from Figures 9 and 10. However, at higher weights, setting fines will result in an excessive burden on renewable energy consumption entities. Therefore, when promoting the consumption policy of renewable energy electricity, it is necessary to consider the weight and penalty together.

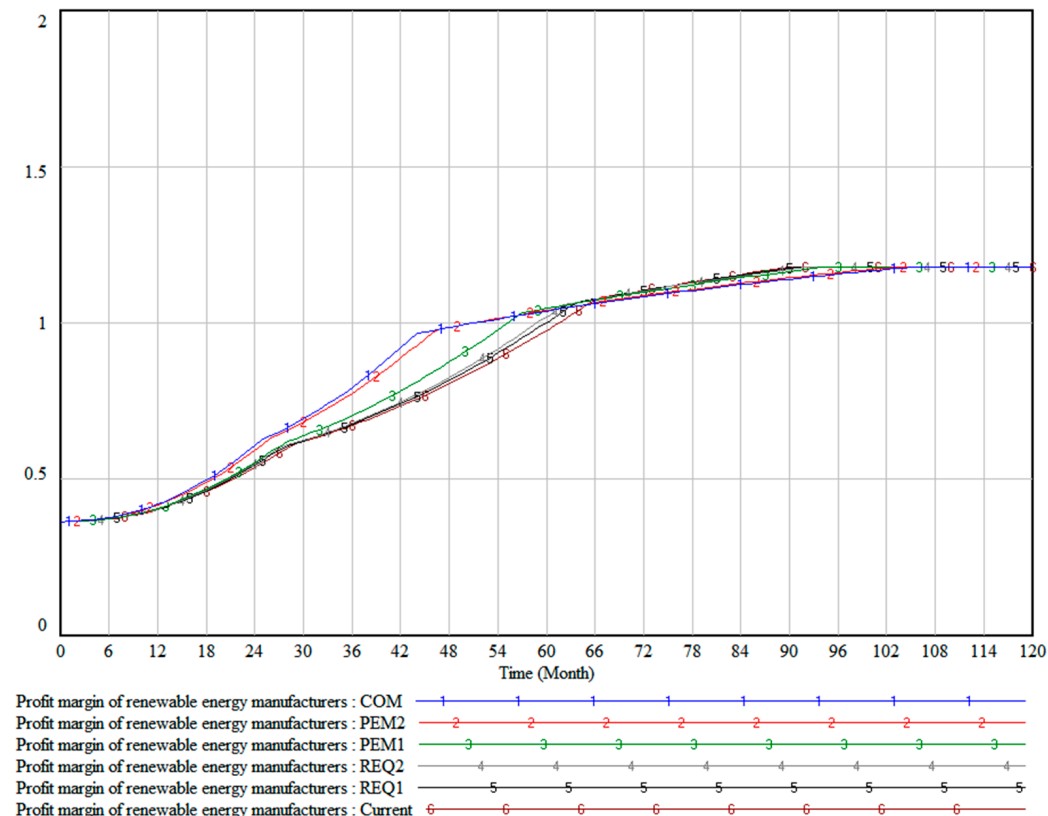

**Figure 9.** Profit space of renewable energy manufacturers in a comprehensive scenario.

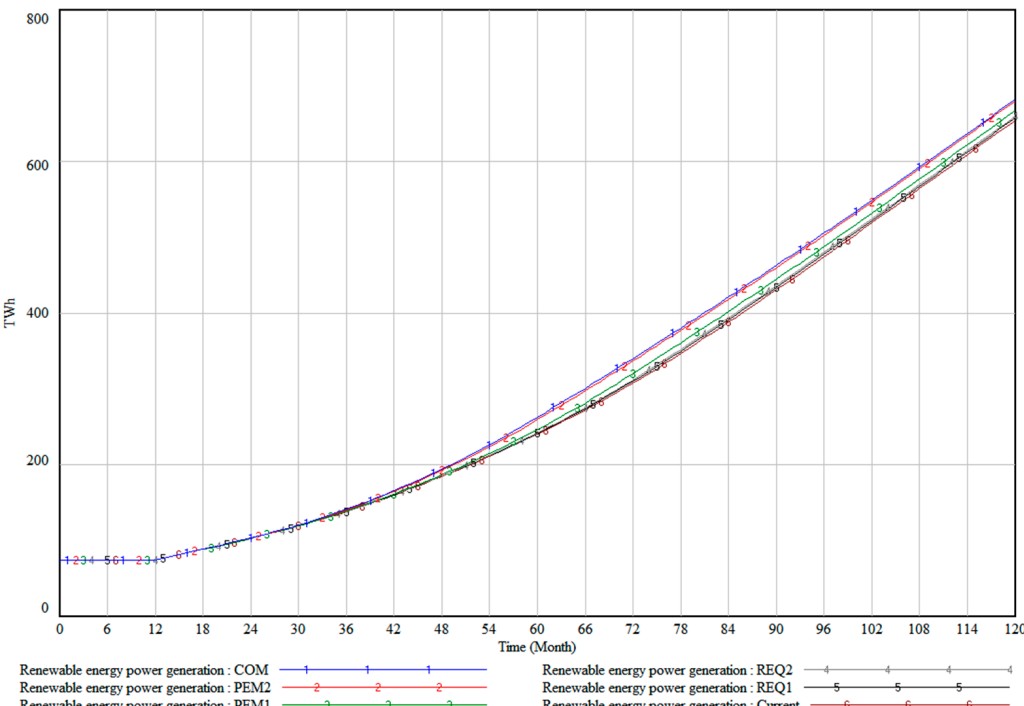

**Figure 10.** Renewable energy power generation under comprehensive scenario.

## 6. Main Conclusions

Based on the analysis of the operation mechanism of China's renewable energy consumption guarantee mechanism, this paper constructs a system dynamics model of a renewable energy consumption guarantee mechanism, and simulates the trend of the renewable energy consumption market. Based on the analysis of the above results, the main conclusions are as follows:

Firstly, according to the results in Section 5.1, it was found that as the weight of renewable-energy-electricity consumption increases, the rate of increase in green-certificate prices increases, from 0.3 yuan/kWh to 0.6 yuan/kWh, but the impact on excess-consumption prices is not significant. In order to achieve the goal of carbon emission reduction, the improvement of power supply structure has become an inevitable trend. The willingness to subscribe for green certificates will be significantly improved with the promotion of the assessment of the responsibility weight of renewable energy power consumption. Jifeng Li et al. (2023) also found that an increase in the proportion of green certificate quotas will gradually lead to the transition of electricity production towards renewable energy, which is consistent with the results obtained in this study [50].

Secondly, according to the results in Section 5.2, it was found that punishing those who fail to fulfill their consumption responsibilities promotes the increase in green-certificate prices, excess-consumption prices, and renewable energy installed capacity. With the strengthening of punishment, the time when the green-card price reached its maximum value significantly advanced, from 2026 to 2025 and then to 2024. The introduction of a punishment mechanism will enhance the constraint of consumer entities, increasing their demand for green certificates and excess consumption, and driving up their prices under the influence of market mechanisms. Wenhui Zhao et al. (2022) believe that the penalty coefficient has a more intuitive impact on the profits of e-commerce merchants. As the penalty coefficient increases, the profits of e-commerce merchants decrease [51]. It can be seen that the punishment mechanism is also the reason for promoting e-commerce to increase their demand for green electricity.

Thirdly, according to the content analysis of Section 5.3, the renewable energy consumption weight and the punishment mechanism of the consumption market are comprehensively considered in the renewable energy power consumption system model, which can promote the profit space of renewable energy manufacturers, and then increase the green power generation. Comprehensively considering the renewable energy power generation companies under the market improvement measures is more conducive to the increase in renewable energy power generation, and thus promotes the improvement of the power structure. In order to achieve the task of renewable energy power consumption, it is necessary to effectively coordinate the improvement measures of the consumption market. Jianlei Mo et al., in 2018, stated that China needs to comprehensively consider multiple measures to ensure the achievement of emission reduction targets [52]. Therefore, in order to achieve the development goals of renewable energy in the Beijing–Tianjin–Hebei region, multiple measures need to be reasonably matched.

The model and conclusions constructed in this article on promoting the consumption of renewable energy have reference value for the green electricity consumption of other provinces in China. At the same time, the main conclusions of this article can be used as a reference for other countries around the world to improve their renewable energy consumption policies. Finally, there are some limitations in this paper that need to be further expanded and improved in future research. On the one hand, in addition to the guarantee mechanism for renewable energy power consumption, China has other measures to promote renewable energy consumption, such as government subsidies, tax incentives, international cooperation and so on. On the other hand, the ultimate goal of the renewable energy consumption policy is to realize China's "double carbon" goal. In order to realize the emission reduction goal of the power industry, China has implemented low-carbon measures such as energy trading policies, new energy storage technologies, and clean and efficient coal technologies. In order to achieve the policy effectiveness, synergistic

development among multiple policies can be considered to scale policy conflicts and redundancy. However, this paper does not fully consider the above problems, which are issues that can be fully considered in our future research work.

**Author Contributions:** Conceptualization, Y.L. (Yan Lu) and X.L.; methodology, H.L.; investigation, H.W.; resources, J.K. and C.Z.; writing—original draft preparation, M.C. and Y.L. (Yan Li); writing—review and editing, X.S.; visualization, J.X.; and supervision, T.F. All authors have read and agreed to the published version of the manuscript.

**Funding:** This paper is supported by Technical Innovation Cost Projects 'Research on the market trading mechanism of Beijing-Tianjin-Hebei new power system under carbon peaking and carbon neutrality goals (Grant No. SGJBJY00JJJS2200020)' funded by State Grid Jibei Electric Power Company Limited Economic Research Institute; Technical Innovation Cost Project 'Research on the effect of green power market trading mechanism on electricity market operation (Grant No. SGJBJY00JJJS2310036)' funded by State Grid Jibei Electric Power Company Limited Economic Research Institute. the National Natural Science Foundation of China (Grant No. 42171278), the National Natural Science Foundation of China (Grant No. 71991481 and No. 71991480), and the Fundamental Research Funds for the Central Universities (Grant No. 2652019083).

**Data Availability Statement:** Not applicable.

**Acknowledgments:** The authors are grateful to the editors and reviewers for their helpful comments and suggestions.

**Conflicts of Interest:** The authors declare that they have no known competing financial interests or personal relationships that could have appeared to influence the work reported in this paper.

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
