# Peer review of "What Is the Impact of the Renewable Energy Power Absorption Guarantee Mechanism on China’s Green Electricity Market?"

_energies, doi:10.3390/en16217434_

Round 1
Reviewer 1 Report
Comments and Suggestions for Authors
This article discusses topical and important issues related to the development of an energy efficient policy in the field of the use of renewable energy sources and the optimization of relevant market mechanisms.
Among the main external and internal factors that affect the potential market profitability and, consequently, the attractiveness of renewable energy projects, experts highlight the following: 1) measures and quality of state support for the development of renewable energy sources; 2) financial performance of renewable energy companies; 3) the price at which the generated electricity can be sold; 4) the attractiveness of a particular country, etc. A significant number of such indicators, as well as the lack of consensus among experts, contributed to setting the goal of the study - a comprehensive study of the factors that affect the investment attractiveness of companies and renewable energy projects from the standpoint of a market profitability indicator. To achieve this goal, the following tasks are supposed to be solved: an integral analysis of investment activity in the global renewable energy market; quantitative assessment of financial indicators affecting the development of renewable energy companies (on the example of sector leaders), analysis of the reliability of the obtained profitability models for companies in the sector; expert assessment of the level of influence of specific risks on the performance of companies (on the example of the leading countries in the sector), analysis of the relevance of the results obtained; comparison of the received results with the data of world researches.
The results obtained in the article are of undoubted interest to readers in the field under consideration.
However, there are the following issues that should be clarified:
1. In section “2.1. Implementation effects of Renewable energy consumption guarantee mechanism” should expand the literature review in the field of considering the use of renewable energy sources in various regions of the world, including an analysis of their advantages and disadvantages over traditional energy sources. This is necessary for the development of a systematic concept for managing the renewable energy market, which is of particular relevance due to the increase in electricity consumption in the world. In this regard, the following works could be considered: https://doi.org/10.1016/B978-0-12-819727-1.00117-5 (hydropower), https://doi.org/10.1109/FarEastCon.2019.8934222 (solar power), https://doi.org/10.1016/j.renene.2019.05.117 (wind and wave energy).
2. The section "Materials and methods" should be added to the article.
3. As part of the research, it was possible to conduct a comprehensive analysis of investment processes in the global renewable energy market. It includes a regional and institutional assessment of the market, a study of the pace of public and private investment in the sector, including in the context of types of renewable energy, a rating of countries for the commissioning of new capacities, etc. It was possible to add the relevant results to the section “Operation mechanism of renewable energy electricity consumption guarantee 202 mechanism".
4. It is necessary to dwell in more detail on the research methods that were used in the formation of the scheme presented in Figure d. Were the methods of expert assessments, cluster analysis, etc. used in this case? to obtain weighted averages of the fuzzy number of parameters considered and highlight the most significant factors influencing the renewable energy market?
5. A more complete analysis of the stock flow pattern shown in Figure 2 should be provided. How applicable is it under similar conditions in different regions of the world?
6. It is necessary to increase the letter designations of the curves shown in Figures 9, 10.
7. According to the data presented in Figures 3-7, a regression analysis should be carried out to obtain mathematical models for calculating and forecasting the production and consumption of electricity using renewable energy sources.
8. It is not entirely clear from the article whether the forecasting of electricity generation was carried out through renewable energy? It is necessary to dwell on methods for predicting electricity, primarily based on the use of machine learning methods, in particular, neural networks and hybrid forecasting methods, which can be seen from the analysis of the work: https://doi.org/10.21177/1998-4502-2022 -14-3-486-493.
9. Specific numerical practical results obtained in the work should be added to the article for their implementation in production at existing facilities.
10. In the conclusions, one should dwell in more detail on the testing of the results obtained and the prospects for applying the developed methodology for studying the renewable energy market in various regions of the world.
Author Response
Dear Reviewer,
RE: Manuscript ID: energies-2610469
Thanks very much for your efficient and meticulous work. We would like to thank Energies for giving us the opportunity to revise our manuscript. And we are very keen to publish our work in this journal.
We thank the reviewers for their careful read and thoughtful comments on previous draft. They are reasonable and clear to show us the defects of our paper. We accept all the comments. According to the comments, we have revised our manuscript carefully.
Below is our response to the reviewers’ comments. Thanks again for your kind suggestions. If you have any questions, please feel free to contact us, we will make efforts to do better. Thanks for all your help.
Best regards,
Haoran Wang (Corresponding Author)
Reviewer #1-- To make it easier to find, we have marked the revised contents in blue color like this in the revised version.
- In section “2.1.Implementation effects of Renewable energy consumption guarantee mechanism” should expand the literature review in the field of considering the use of renewable energy sources in various regions of the world, including an analysis of their advantages and disadvantages over traditional energy sources. This is necessary for the development of a systematic concept for managing the renewable energy market, which is of particular relevance due to the increase in electricity consumption in the world. In this regard, the following works could be considered: https://doi.org/10.1016/B978-0-12-819727-1.00117-5 (hydropower), https://doi.org/10.1109/FarEastCon.2019.8934222 (solar power), https://doi.org/10.1016/j.renene.2019.05.117 (wind and wave energy).
Response: According to the reviewer’s comments, we have added Section “2.1 The consumption of renewable energy in the world” to analyze the advantages of renewable energy consumption and renewable energy in other countries around the world. In addition, we have focused on the references you recommended and cited them in the article. Please see Line 98~Line 143 on Page 3.
- The section "Materials and methods" should be added to the article.
Response: According to the reviewer’s comments, We have adjusted the framework of the article and added Section "4 Materials and methods". In Section 4, we supplemented an introduction to the system dynamics approach, and added important formulas in the model regarding the excess consumption trading market (include green certificate) and the green electricity market. Please see Line 331 on Page 7~Line 401 on Page 10.
- As part of the research, it was possible to conduct a comprehensive analysis of investment processes in the global renewable energy market.It includes a regional and institutional assessment of the market, a study of the pace of public and private investment in the sector, including in the context of types of renewable energy, a rating of countries for the commissioning of new capacities, etc. It was possible to add the relevant results to the section “Operation mechanism of renewable energy electricity consumption guarantee 202 mechanism".
Response: According to the reviewer’s comments, we added analysis in the Section 3. On the one hand, analysis of global renewable energy investment and its investment types; On the other hand, China's investment in renewable energy generates an evaluation of new production capacity. Please see Line 253~Line 278 on Page 6.
- It is necessary to dwell in more detail on the research methods that were used in the formation of the scheme presented in Figure d.Were the methods of expert assessments, cluster analysis, etc. used in this case? to obtain weighted averages of the fuzzy number of parameters considered and highlight the most significant factors influencing the renewable energy market?
Response: According to the reviewer’s comments, We have made modifications to Figure 1 to illustrate the relationship between multiple subjects, as shown in Figure 1. in addition, we have added an explanation of the method we used. Please see Figure 1, Line 331 on Page 7~Line 401 on Page 10.
- A more complete analysis of the stock flow pattern shown in Figure 2 should be provided. How applicable is it under similar conditions in different regions of the world?
Response: According to the reviewer’s comments, We have added a complete description of Figure 2. On the one hand, explain the basis for model construction. On the other hand, explain the applicability of this model in other provinces in China. Please see Line 363~Line 376 on Page 8.
- It is necessary to increase the letter designations of the curves shown in Figures 9, 10.
Response: According to the reviewer’s comments, we added the letter designations of the curves shown in Figures 9, 10. Please see Figures 9, 10 on Page 16 and 17.
- According to the data presented in Figures 3-7, a regression analysis should be carried out to obtain mathematical models for calculating and forecasting the production and consumption of electricity using renewable energy sources.
Response: According to the reviewer’s comments, we have added an analysis of the results in Figures 3-10 in the Section 5. At the same time, we have added the calculation formula for key variables in the Section 4. This paper uses the method of system dynamics to simulate the model. Please see Line 475~Line 483 on Page 13, Line 539~Line 544 on Page 15, Line 570~Line 574 on Page 17.
- It is not entirely clear from the article whether the forecasting of electricity generation was carried out through renewable energy?It is necessary to dwell on methods for predicting electricity, primarily based on the use of machine learning methods, in particular, neural networks and hybrid forecasting methods, which can be seen from the analysis of the work: https://doi.org/10.21177/1998-4502-2022 -14-3-486-493.
Response: According to the reviewer’s comments, we have added the calculation formula for key variables in the Section 4.3. Since system dynamics (SD) is a subject that studies the dynamic complexity of system, it is mainly used to study the structure and function of complex system and the interdependence of dynamic behaviors. This method can analyze the causal relationship and feedback mechanism among decision making units from micro perspective, and can predict the future dynamic behaviors. Therefore, this section uses the SD model to stimulate the impact of renewable energy consumption guarantee mechanism on the renewable energy electricity market. Please see Line 380 on Page 9~Line 401 on Page 10.
- Specific numerical practical results obtained in the work should be added to the article for their implementation in production at existing facilities.
Response: According to the reviewer’s comments, We have added Section “4.4 Data source and related parameter description” to explain the parameter settings and basis in the model. Please see Line 402~Line 412 on Page 10.
- In the conclusions, one should dwell in more detail on the testing of the results obtained and the prospects for applying the developed methodology for studying the renewable energy market in various regions of the world.
Response: According to the reviewer’s comments, we have added explanations to the main conclusions obtained in this study from three aspects. We compared the conclusions of this study with the achievements of other scholars. In addition, the model developed in this article and the main conclusions drawn illustrate the practicality of its role in other provinces of China and other regions of the world. Please see Line 581 on Page 17~Line 622 on Page 18.
From the above, we have marked all the revised and added contents in the blue color.
Finally, thanks again for your efficient and meticulous work.

Reviewer 2 Report
Comments and Suggestions for Authors
Suggested changes for authors are attached in the file below.

Author Response
Dear Reviewer,
RE: Manuscript ID: energies-2610469
Thanks very much for your efficient and meticulous work. We would like to thank Energies for giving us the opportunity to revise our manuscript. And we are very keen to publish our work in this journal.
We thank the reviewers for their careful read and thoughtful comments on previous draft. They are reasonable and clear to show us the defects of our paper. We accept all the comments. According to the comments, we have revised our manuscript carefully.
Below is our response to the reviewers’ comments. Thanks again for your kind suggestions. If you have any questions, please feel free to contact us, we will make efforts to do better. Thanks for all your help.
Best regards,
Haoran Wang (Corresponding Author)
Reviewer #2-- To make it easier to find, we have marked the revised contents in yellow color like this in the revised version.
1.Too narrow a literature review. This study provides theoretical references for formulating clean and low-emission policy goals in the energy sector and should be solidly embedded in the literature review.
Response: According to the reviewer’s comments, we have added Section “2.1 The consumption of renewable energy in the world” to analyze the advantages of renewable energy consumption and renewable energy in other countries around the world. In addition, we have focused on the references you recommended and cited them in the article. Please see Line 98~Line 143 on Page 3.
2.The reference year (2021) and time frame of the study must be justified.
Response: According to the reviewer’s comments, we have added section “4.4 Data source and related parameter description” to explain the parameter settings and basis in the model. Please see Line 402~Line 412 on Page 10.
- I recommend justifying the assumptions of the scenarios, indicating the probability of a given scenario or the most optimistic and pessimistic variant.
Response: According to the reviewer’s comments, we added an explanation of the reasons for setting the scenarios. This paper sets up multiple scenarios for simulation to study the future impact of the reform measures for the renewable power consumption guarantee mechanism. Please see Line 422~Line 429 on Page 10.
4.No practical conclusions from scenario analyses. In the analysis of individual variants, it is worth indicating what impact it will have on the economy, the environment and the standard of living.
Response: According to the reviewer’s comments, we have added an analysis of scenario simulation results in Section 5. Please see Line 475~Line 483 on Page 13, Line 539~Line 544 on Page 15, Line 570~Line 574 on Page 17.
5.It is worth raising the question of who is the real beneficiary of changes in energy certificate prices, is it not the person issuing the fines? As a result, whether the company is able to meet the imposed standards. Is this system good or will there be a situation where standards will be constantly raised for economic benefit?
Response: According to the reviewer’s comments, we added analysis in Section “3 Operation mechanism of renewable energy electricity consumption guarantee mechanism”. The direct beneficiaries of renewable energy certificates are renewable energy generators, and rising prices will result in more green profits. The government, as a regulatory authority, only receives the benefits of punishment. Please see Line 253~Line 278 on Page 6.
6.The conclusions from the research are very poor. The author focuses only on the mechanism of penalties that can promote the profit space of renewable energy producers.
Response: According to the reviewer’s comments, we have added explanations to the main conclusions obtained in this study from three aspects. We compared the conclusions of this study with the achievements of other scholars. In addition, the model developed in this article and the main conclusions drawn illustrate the practicality of its role in other provinces of China and other regions of the world. Please see Line 581 on Page 17~Line 622 on Page 18.
7.Recommendation for the use of factor analysis. When we operate on a data set containing many variables, this method will allow us to identify and aggregate factors that have a potential impact on factors promoting the development of green energy. This will allow for a precise determination of the share of variables influencing the selection of factors and, consequently, actions on the Chinese green energy market.
Response: According to the reviewer’s comments, we have added specific instructions for the method in Section 4. Since system dynamics (SD) is a subject that studies the dynamic complexity of system, it is mainly used to study the structure and function of complex system and the interdependence of dynamic behaviors. This method can analyze the causal relationship and feedback mechanism among decision making units from micro perspective, and can predict the future dynamic behaviors. Therefore, this section uses the SD model to stimulate the impact of renewable energy consumption guarantee mechanism on the renewable energy electricity market. Please see Line 331 on Page 7~Line 401 on Page 10.
From the above, we have marked all the revised and added contents in the yellow color.
Finally, thanks again for your efficient and meticulous work.

Reviewer 3 Report
Comments and Suggestions for Authors
The paper focus on the impact of Renewable Energy Power Absorption Guarantee Mechanism on China's Green Electricity Market, it is very interesting and value. And my suggestions are as follows.
1. In section 2, it is suggested to highlight the innovations and contributions of this paper.
2. In Figure 1, it will be better to show the relationship of green certificate and electricity from renewable energy.
3. In section 6, the detailed measurement of improving policies/market in this paper should be expressed in detail to promote the achievement of carbon peaking and carbon neutrality.
4. The limitation or prospect are suggested to add in the paper.

Minor editing of English language required
Author Response
Dear Reviewer,
RE: Manuscript ID: energies-2610469
Thanks very much for your efficient and meticulous work. We would like to thank Energies for giving us the opportunity to revise our manuscript. And we are very keen to publish our work in this journal.
We thank the reviewers for their careful read and thoughtful comments on previous draft. They are reasonable and clear to show us the defects of our paper. We accept all the comments. According to the comments, we have revised our manuscript carefully.
Below is our response to the reviewers’ comments. Thanks again for your kind suggestions. If you have any questions, please feel free to contact us, we will make efforts to do better. Thanks for all your help.
Best regards,
Haoran Wang (Corresponding Author)
Reviewer #3-- To make it easier to find, we have marked the revised contents in green color like this in the revised version.
- In section 2, it is suggested to highlight the innovations and contributions of this paper.
Response: According to the reviewer’s comments, we have added two innovative points in the “Introduction” section of this article. Please see Line 88~Line 96 on Page 2.
2.In Figure 1, it will be better to show the relationship of green certificate and electricity from renewable energy.
Response: According to the reviewer’s comments, We have made modifications to Figure 1 to illustrate the relationship between multiple subjects, as shown in Figure 1. in addition, we have added an explanation. Please see Figure 1.
3.In section 6, the detailed measurement of improving policies/market in this paper should be expressed in detail to promote the achievement of carbon peaking and carbon neutrality.
Response: According to the reviewer’s comments, we have added explanations to the main conclusions obtained in this study from three aspects. We compared the conclusions of this study with the achievements of other scholars. In addition, the model developed in this article and the main conclusions drawn illustrate the practicality of its role in other provinces of China and other regions of the world. Please see Line 581 on Page 17~Line 622 on Page 18.
- 4. The limitation or prospect are suggested to add in the paper.
Response: According to the reviewer’s comments, we have added a limitations and outlook section to the final part of the article. Limitations and future expectations are presented in terms of policy comprehensiveness and policy synergies to promote renewable energy consumption. Please see Line 622~Line 634 on Page 18.
From the above, we have marked all the revised and added contents in the green color.
Finally, thanks again for your efficient and meticulous work.

Round 2
Reviewer 1 Report
Comments and Suggestions for Authors
The authors' comments have been corrected. I recommend the article for publication.